# Dynamics of SGD with Stochastic Polyak Stepsizes: Truly Adaptive Variants and Convergence to Exact Solution

**Antonio Orvieto**[*]
Department of Computer Science,
ETH Zürich

**Simon Lacoste-Julien**[†]
Mila and DIRO,
Université de Montréal

**Nicolas Loizou**
AMS and MINDS,
Johns Hopkins University

## Abstract

Recently Loizou et al. [22], proposed and analyzed stochastic gradient descent (SGD) with stochastic Polyak stepsize (SPS). The proposed SPS comes with strong convergence guarantees and competitive performance; however, it has two main drawbacks when it is used in non-over-parameterized regimes: (i) It requires a priori knowledge of the optimal mini-batch losses, which are not available when the interpolation condition is not satisfied (e.g., regularized objectives), and (ii) it guarantees convergence only to a neighborhood of the solution. In this work, we study the dynamics and the convergence properties of SGD equipped with new variants of the stochastic Polyak stepsize and provide solutions to both drawbacks of the original SPS. We first show that a simple modification of the original SPS that uses lower bounds instead of the optimal function values can directly solve issue (i). On the other hand, solving issue (ii) turns out to be more challenging and leads us to valuable insights into the method's behavior. We show that if interpolation is not satisfied, the correlation between SPS and stochastic gradients introduces a bias, which effectively distorts the expectation of the gradient signal near minimizers, leading to non-convergence - even if the stepsize is scaled down during training. To fix this issue, we propose DecSPS, a novel modification of SPS, which guarantees convergence to the exact minimizer - without a priori knowledge of the problem parameters. For strongly-convex optimization problems, DecSPS is the first stochastic adaptive optimization method that converges to the exact solution without restrictive assumptions like bounded iterates/gradients.

## 1 Introduction

We consider the stochastic optimization problem:

$$\min_{x \in \mathbb{R}^d} \left[ f(x) = \frac{1}{n} \sum_{i=1}^{n} f_i(x) \right], \tag{1}$$

where each $f_i$ is convex and lower bounded. We denote by $\mathcal{X}^*$ the non-empty set of optimal points $x^*$ of equation (1). We set $f^* := \min_{x \in \mathbb{R}^d} f(x)$, and $f_i^* := \inf_{x \in \mathbb{R}^d} f_i(x)$.

36th Conference on Neural Information Processing Systems (NeurIPS 2022).

---

[*]Corresponding author: `antonio.orvieto@inf.ethz.ch`. Part of this work was done while interning at Mila, Université de Montréal under the supervision of Nicolas Loizou and Simon Lacoste-Julien.

[†]Canada CIFAR AI Chair

In this setting, the algorithm of choice is often Stochastic Gradient Descent (SGD), i.e. $x^{k+1} = x^k - \gamma_k \nabla f_{\mathcal{S}_k}(x^k)$, where $\gamma_k > 0$ is the stepsize at iteration $k$, $\mathcal{S}_k \subseteq [n]$ a random subset of datapoints (minibatch) with cardinality $B$ sampled independently at each iteration $k$, and $\nabla f_{\mathcal{S}_k}(x) := \frac{1}{B} \sum_{i \in \mathcal{S}_k} \nabla f_i(x)$ is the minibatch gradient.

A careful choice of $\gamma_k$ is crucial for most applications [4, 14]. The simplest option is to pick $\gamma_k$ to be constant over training, with its value inversely proportional to the Lipschitz constant of the gradient. While this choice yields fast convergence to the neighborhood of a minimizer, two main problems arise: (a) the optimal $\gamma$ depends on (often unknown) problem parameters — hence often requires heavy tuning ; and (b) it cannot be guaranteed that $\mathcal{X}^*$ is reached in the limit [13, 16, 15]. A simple fix for the last problem is to allow polynomially decreasing stepsizes (second option) [23]: this choice for $\gamma_k$ often leads to convergence to $\mathcal{X}^*$, but hurts the overall algorithm speed. The third option, which became very popular with the rise of deep learning, is to implement an *adaptive* stepsize. These methods do not commit to a fixed schedule, but instead use the optimization statistics (e.g. gradient history, cost history) to tune the value of $\gamma_k$ at each iteration. These stepsizes are known to work very well in deep learning [35], and include Adam [19], Adagrad [11], and RMSprop [29].

Ideally, a theoretically grounded adaptive method should yield fast convergence to $\mathcal{X}^*$ without knowledge of problem dependent parameters, such as the gradient Lipshitz constant or the strong convexity constant. As a result, an ideal adaptive method should require very little tuning by the user, while matching the performance of a fine-tuned $\gamma_k$. However, while in practice this is the case for common adaptive methods such as Adam and AdaGrad, the associated convergence rates often rely on strong assumptions — e.g. that the iterates live on a bounded domain, or that gradients are uniformly bounded in norm [11, 32, 31]. While the above assumptions are valid in the constrained setting, they are problematic for problems defined in the whole $\mathbb{R}^d$.

A promising new direction in the adaptive stepsizes literature is based on the idea of Polyak stepsizes, introduced by [25] in the context of deterministic convex optimization. Recently [22] successfully adapted Polyak stepsizes to the stochastic setting, and provided convergence rates matching fine-tuned SGD — while the algorithm does not require knowledge of the unknown quantities such as the gradient Lipschitz constant. The results especially shines in the overparameterized strongly convex setting, where linear convergence to $x^*$ is shown. This result is especially important since, under the same assumption, no such rate exists for AdaGrad (see e.g. [31] for the latest results) or other adaptive stepsizes. Moreover, the method was shown to work surprisingly well on deep learning problems, without requiring heavy tuning [22].

Even if the stochastic Polyak stepsize (SPS) [22] comes with strong convergence guarantees, it has two main drawbacks when it is used in non-over-parameterized regimes: (i) It requires *a priori* knowledge of the optimal mini-batch losses, which are not often available for big batch sizes or regularized objectives (see discussion in §1.1) and (ii) it guarantees convergence only to a neighborhood of the solution. In this work, we study the dynamics and the convergence properties of SGD equipped with new variants of SPS for solving general convex optimization problems. Our new proposed variants provide solutions to both drawbacks of the original SPS.

## 1.1 Background and Technical Preliminaries

The stepsize proposed by [22] is

$$\gamma_k = \min \left\{ \frac{f_{\mathcal{S}_k}(x^k) - f_{\mathcal{S}_k}^*}{c \|\nabla f_{\mathcal{S}_k}(x^k)\|^2}, \gamma_b \right\}, \tag{SPS$_{\max}$}$$

where $\gamma_b, c > 0$ are problem-independent constants, $f_{\mathcal{S}_k} := \frac{1}{|\mathcal{S}_k|} \sum_{i \in \mathcal{S}_k} f_i$, $f_{\mathcal{S}_k}^* = \inf_{x \in \mathbb{R}^d} f_{\mathcal{S}_k}(x)$.

**Dependency on $f_{\mathcal{S}_k}^*$.** Crucially the algorithm requires knowledge of $f_{\mathcal{S}_k}^*$ for every realization of the mini-batch $\mathcal{S}_k$. In the non-regularized overparametrized setting (e.g. neural networks), $f_{\mathcal{S}_k}$ is often zero for every subset $\mathcal{S}$ [34]. However, this is not the only setting where $f_{\mathcal{S}}^*$ is computable: e.g., in the regularized logistic loss with batch size 1, it is possible to recover a cheap closed form

expression for each $f_i^*$ [22]. Unfortunately, if the batch-size is bigger than $1$ or the loss becomes more demanding (e.g. cross-entropy), then *no such closed-form computation is possible*.

**Rates and comparison with AdaGrad.** In the convex overparametrized setting (more precisely, under the interpolation condition, i.e. $\exists\, x^* \in \mathcal{X}^* : \inf_{x \in \mathbb{R}^d} f_{\mathcal{S}}(x) = f_{\mathcal{S}}(x^*)$ for all $\mathcal{S}$, see also §2), $\text{SPS}_{\max}$ enjoys a convergence speed of $\mathcal{O}(1/k)$, without requiring knowledge of the gradient Lipschitz constant or other problem parameters. Recently, [31] showed that the same rate can be achieved for AdaGrad in the same setting. However, there is an important difference: the rate of [31] is technically $\mathcal{O}(dD^2/k)$, where $d$ is the problem dimension and $D^2$ is a global bound on the squared distance to the minimizer, which is assumed to be finite. Not only does $\text{SPS}_{\max}$ not have this dimension dependency, which dates back to crucial arguments in the AdaGrad literature [11, 21], but also does not require bounded iterates. While this assumption is satisfied in the constrained setting, it has no reason to hold in the unconstrained scenario. Unfortunately, this is a common problem of all AdaGrad variants: with the exception of [33] (which works in a slightly different scenario), no rate can be provided in the stochastic setting without the bounded iterates/gradients [12] assumption — even after assuming strong convexity. However, in the non-interpolated setting, AdaGrad enjoys a convergence guarantee of $\mathcal{O}(1/\sqrt{k})$ (with the bounded iterates assumption). A similar rate does not yet exist for SPS, and our work aims at filling this gap.

## 1.2 Main Contributions

As we already mentioned, in the non-interpolated setting $\text{SPS}_{\max}$ has the following issues:

**Issue (1)**: For $B > 1$ (minibatch setting), $\text{SPS}_{\max}$ requires the exact knowledge of $f_{\mathcal{S}}^*$. This is not practical.

**Issue (2)**: $\text{SPS}_{\max}$ guarantees convergence to a neighborhood of the solution. It is not clear how to modify it to yield convergence to the exact minimizer.

Having the above two issues in mind, the main contributions of our work (see also Table 1 for a summary of the main complexity results obtained in this paper) are summarized as follows:

- In §3, we provide a direct solution for Issue (1). We explain how only a lower bound on $f_{\mathcal{S}}^*$ (trivial if all $f_i$s are non-negative) is required for convergence to a neighborhood of the solution. While this neighborhood is bigger that the one for $\text{SPS}_{\max}$, our modified version provides a practical baseline for the solution to the second issue.

- We explain why Issue (2) is highly non-trivial and requires an in-depth study of the bias induced by the interaction between gradients and Polyak stepsizes. Namely, we show that simply multiplying the stepsize of $\text{SPS}_{\max}$ by $1/\sqrt{k}$ — which would work for vanilla SGD [23] — yields a bias in the solution found by SPS (§4), regardless of the estimation of $f_{\mathcal{S}}^*$.

- In §5, we provide a solution to the problem (Issue (2)) by introducing additional structure — as well as the fix to Issue (1) — into the stepsize. We call the new algorithm *Decreasing SPS* (DecSPS), and provide a convergence guarantee under the bounded domain assumption — matching the standard AdaGrad results.

- In §5.2 we go one step further and show that, if strong convexity is assumed, iterates are bounded with probability 1 and hence we can remove the bounded iterates assumption. To the best of our knowledge, DecSPS, is the first stochastic adaptive optimization method that converges to the exact solution without assuming strong assumptions like bounded iterates/gradients.

- In §5.3 we provide extensions of our approach to the non-smooth setting.

- In §6, we corroborate our theoretical results with experimental testing.

## 2 Background on Stochastic Polyak Stepsize

In this section, we provide a concise overview of the results in [22], and highlight the main assumptions and open questions.

To start, we remind the reader that problem (1) is said to be interpolated if there exists a problem solution $x^* \in \mathcal{X}^*$ such that $\inf_{x \in \mathbb{R}^d} f_i(x) = f_i(x^*)$ for all $i \in [n]$. The degree of interpolation

| Stepsize | Citation | Assumptions | No Knowledge of $f_{\mathcal{S}}^*$ | Exact Convergence | Theorem |
|---|---|---|---|---|---|
| SPS$_{\max}$ | [22] | convex, smooth | ✗ | ✗ | Thm. 1 |
| SPS$_{\max}^{\ell}$ | This paper | convex, smooth | ✓ | ✗ | Thm. 2, Cor. 1 |
| DecSPS | This paper | convex, smooth, bounded iterates | ✓ | ✓ | Cor. 2, $\mathcal{O}(1/\sqrt{K})$ |
| | This paper | strongly-convex, smooth | ✓ | ✓ | Thm. 4, $\mathcal{O}(1/\sqrt{K})$ |
| DecSPS-NS | This paper | convex, bounded iterates/grads | ✓ | ✓ | Cor. 3, $\mathcal{O}(1/\sqrt{K})$ |

Table 1: Summary of the considered stepsizes and the corresponding theoretical results in the non-interpolated setting. The studied quantity in all Theorems, with respect to which all rates are expressed is $\mathbb{E}\left[f(\bar{x}^K) - f(x^*)\right]$, where $\bar{x}^K = \frac{1}{K}\sum_{k=0}^{K-1} x^k$. In addition, for all converging methods, we consider the stepsize scaling factor $c_k = \mathcal{O}(\sqrt{k})$, formally defined in the corresponding sections. For the methods without exact convergence, we show in §4 that any different scaling factor cannot make the algorithm convergent.

at batch size $B$ can be quantified by the following quantity, introduced by [22] and studied also in [31, 9]: fix a batch size $B$, and let $\mathcal{S} \subseteq [n]$ with $|\mathcal{S}| = B$.

$$\sigma_B^2 := \mathbb{E}_{\mathcal{S}}[f_{\mathcal{S}}(x^*) - f_{\mathcal{S}}^*] = f(x^*) - \mathbb{E}_{\mathcal{S}}[f_{\mathcal{S}}^*] \tag{2}$$

It is easy to realize that as soon as problem (1) is interpolated, then $\sigma_B^2 = 0$ for each $B \leq n$. In addition, note that $\sigma_B^2$ is non-increasing as a function of $B$.

We now comment on the main result from [22].

**Theorem 1** (Main result of [22]). *Let each $f_i$ be $L_i$-smooth convex functions. Then SGD with* SPS$_{\max}$*, mini-batch size $B$, and $c = 1$, converges as:* $\mathbb{E}\left[f(\bar{x}^K) - f(x^*)\right] \leq \frac{\|x^0 - x^*\|^2}{\alpha K} + \frac{2\gamma_b \sigma_B^2}{\alpha}$, *where $\alpha = \min\left\{\frac{1}{2cL_{\max}}, \gamma_b\right\}$ and $\bar{x}^K = \frac{1}{K}\sum_{k=0}^{K-1} x^k$. If in addition $f$ is $\mu$-strongly convex, then, for any $c \geq 1/2$, SGD with* SPS$_{\max}$ *converges as:* $\mathbb{E}\|x^k - x^*\|^2 \leq (1 - \mu\alpha)^k \|x^0 - x^*\|^2 + \frac{2\gamma_b \sigma_B^2}{\mu\alpha}$, *where again $\alpha = \min\{\frac{1}{2cL_{\max}}, \gamma_b\}$ and $L_{\max} = \max\{L_i\}_{i=1}^n$ is the maximum smoothness constant.*

In the overparametrized setting, the result guarantees convergence to the exact minimizer, without knowledge of the gradient Lipschitz constant (as vanilla SGD would instead require) and without assuming bounded iterates (in contrast to [31]).

As soon as (1) a *regularizer* is applied to the loss (e.g. $L2$ penalty), or (2) the number of datapoints gets comparable to the dimension, then the problem is not interpolated and SPS$_{\max}$ only converges to a neighborhood and it gets impractical to compute $f_{\mathcal{S}}^*$ — *this is the setting we study in this paper.*
*Remark* 1 (What if $\|\nabla f_{\mathcal{S}_k}\| = 0$?). In the rare case that $\|\nabla f_{S_k}(x^k)\|^2 = 0$, there is no need to evaluate the stepsize. In this scenario, the update direction $\nabla f_{S_k}(x^k) = 0$ and thus the iterate is not updated irrespective of the choice of step-size. If this happens, the user should simply sample a different minibatch. We note that in our experiments (see §6), such event never occurred.

**Related work on Polyak stepsize:** The classical Polyak stepsize [25] has been successfully used in the analysis of deterministic subgradient methods in different settings [5, 7, 18]. First attempts on providing an efficient variant of the stepsize that works well in the stochastic setting were made in [3, 24]. However, as explained in [22], none of these approaches provide a natural stochastic extension with strong theoretical convergence guarantees, and thus Loizou et al. [22] proposed the stochastic Polyak stepsize SPS$_{\max}$ as a better alternative.[3] Despite its recent appearance, SPS$_{\max}$ has already been used and analyzed as a stepsize for SGD for solving structured non-convex problems [15], in combination with other adaptive methods [31], with a moving target [17] and in the update rule of stochastic mirror descent [9]. These extensions are orthogonal to our approach, and we

---

[3] A variant of SGD with SPS$_{\max}$ was also proposed by Asi and Duchi [2] as a special case of a model-based method called the lower-truncated model. Asi and Duchi [2] also proposed a decreasing step-size variant of SPS$_{\max}$ which is closely related but different than the DecSPS that we propose in §5. Among some differences, they assume interpolation for their convergence results whereas we do not in §5. We describe the differences between our work and Asi and Duchi [2] in more detail in Appendix A.

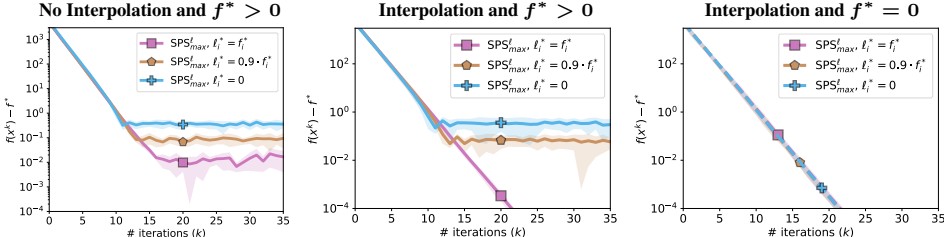

Figure 1: We consider a 100 dim problem with $n = 100$ datapoints where each $f_i = \frac{1}{2}(x - x_i^*)^\top H_i(x - x_i^*) + f_i^*$, with $f_i^* = 1$ for all $i \in [n]$ and $H_i$ a random SPD matrix generated using the standard Gaussian matrix $A_i \in \mathbb{R}^{d \times 3d}$ as $H_i = A_i A_i^\top /(3d)$. If $x_i^* \neq x_j^*$ for $i \neq j$, then the problem does **not satisfy interpolation (left plot)**. Instead, if all $x_i^*$s are equal, **then the problem is interpolated (central plot)**. The plot shows the behaviour of $\text{SPS}^\ell_{\max}$ ($\gamma_b = 2$) for different choices of the approximated suboptimality $\ell_i^*$. We plot (mean and std deviation over 10 runs) the function suboptimality level $f(x) - f(x^*)$ for different values of $\ell_i^*$. Note that, if instead $f_i^* = 0$ for all $i$ then all the shown **algorithms coincide (right plot)** and converge to the solution.

speculate that our proposed variants can also be used in the above settings. We leave such extensions for future work.

## 3  Removing $f_{\mathcal{S}}^*$ from SPS

As motivated in the last sections, computing $f_{\mathcal{S}}^*$ in the non-interpolated setting is not practical. In this section, we explore the effect of using a lower bound $\ell_{\mathcal{S}}^* \leq f_{\mathcal{S}}^*$ instead in the $\text{SPS}_{\max}$ definition.

$$\gamma_k = \min\left\{ \frac{f_{\mathcal{S}_k}(x^k) - \ell_{\mathcal{S}_k}^*}{c\|\nabla f_{\mathcal{S}_k}(x^k)\|^2}, \gamma_b \right\}, \tag{$\text{SPS}^\ell_{\max}$}$$

Such a lower bound is easy to get for many problems of interest: indeed, for standard regularized regression and classification tasks, the loss is non-negative hence one can pick $\ell_{\mathcal{S}}^* = 0$, for any $\mathcal{S} \subseteq [n]$.

The obvious question is: what is the effect of estimating $\ell_{\mathcal{S}}^*$ on the convergence rates in Thm. 1? We found that the proof of [22] is easy to adapt to this case, by using the following fundamental bound (see also Lemma 3): $\frac{1}{2cL_{\mathcal{S}_k}} \leq \frac{f_{\mathcal{S}_k}(x^k) - f_{\mathcal{S}_k}^*}{c\|\nabla f_{\mathcal{S}_k}(x^k)\|^2} \leq \frac{f_{\mathcal{S}_k}(x^k) - \ell_{\mathcal{S}_k}^*}{c\|\nabla f_{\mathcal{S}_k}(x^k)\|^2}$.

The following results can be seen as an easy extension of the main result of [22], under a newly defined suboptimality measure:

$$\hat{\sigma}_B^2 := \mathbb{E}_{\mathcal{S}_k}[f_{\mathcal{S}_k}(x^*) - \ell_{\mathcal{S}_k}^*] = f(x^*) - \mathbb{E}_{\mathcal{S}_k}[\ell_{\mathcal{S}_k}^*]. \tag{3}$$

**Theorem 2.** *Under $\text{SPS}^\ell_{\max}$, the same exact rates in Thm. 1 hold (under the corresponding assumptions), after replacing $\sigma_B^2$ with $\hat{\sigma}_B^2$.*

And we also have an easy practical corollary.

**Corollary 1.** *In the context of Thm. 2, assume all $f_i$s are non-negative and estimate $\ell_{\mathcal{S}}^* = 0$ for all $\mathcal{S} \subseteq [n]$. Then the same exact rates in Thm. 1 hold for $\text{SPS}^\ell_{\max}$, after replacing $\sigma_B^2$ with $f^* = f(x^*)$.*

A numerical illustration of this result can be found in Fig. 1. In essence, both theory and experiments confirm that, if interpolation is not satisfied, then we have a linear rate until a convergence ball, where the size is optimal under exact knowledge of $f_{\mathcal{S}}^*$. Instead, under interpolation, if all the $f_i$s are non-negative and $f^* = 0$, then $\text{SPS}_{\max} = \text{SPS}^\ell_{\max}$. Finally, in the less common case in practice where $f^* > 0$ but we still have interpolation, then $\text{SPS}_{\max}$ converges to the exact solution while $\text{SPS}^\ell_{\max}$ does not. To conclude $\text{SPS}^\ell_{\max}$ does not (of course) work better than $\text{SPS}_{\max}$, but it is a practical variant which we can use as a baseline in §5 for an adaptive stochastic Polyak stepsize with convergence to the true $x^*$ in the non-interpolated setting.

## 4 Bias in the SPS dynamics.

In this section, we study convergence of the standard $\text{SPS}_{\max}$ in the non-interpolated regime, under an additional (decreasing) multiplicative factor, in the most ideal setting: batch size 1, and we have knowledge of each $f_i^*$. That is, we consider $\gamma_k = \min\{\frac{f_{i_k}(x^k)-f_{i_k}^*}{c_k\|\nabla f_{i_k}(x^k)\|^2}, \gamma_b\}$ with $c_k \to \infty$, e.g. $c_k = \mathcal{O}(\sqrt{k})$ or $c_k = \mathcal{O}(k)$. We note that, in the SGD case, simply picking e.g. $\gamma_k = \gamma_0/\sqrt{k+1}$ would guarantee convergence of $f(x^k)$ to $f(x^*)$, in expectation and with high probability [20, 23]. Therefore, it is natural to expect a similar behavior for SPS, if $1/c_k$ safisfies the usual Robbins-Monro conditions [27]: $\sum_{k=0}^{\infty} 1/c_k = \infty, \sum_{k=0}^{\infty} 1/c_k^2 < \infty$.

*We show that this is not the case*: quite interestingly, $f(x^k)$ converges to a biased solution due to the *correlation between* $\nabla f_{i_k}$ *and* $\gamma_k$. we show this formally, in the case of non-interpolation (otherwise both SGD and SPS do not require a decreasing learning rate).

**Counterexample.** Consider the following finite-sum setting: $f(x) = \frac{1}{2}f_1(x) + \frac{1}{2}f_2(x)$ with $f_1(x) = \frac{a_1}{2}(x-1)^2, f_2(x) = \frac{a_2}{2}(x+1)^2$. To make the problem interesting, we choose $a_1 = 2$ and $a_2 = 1$: this introduces asymmetry in the average landscape with respect to the origin. During optimization, we sample $f_1$ and $f_2$ independently and seek convergence to the unique minimizer $x^* = \frac{a_1-a_2}{a_1+a_2} = 1/3$. The first thing we notice is that $x^*$ is not a stationary point for the dynamics under SPS. Indeed note that since $f_i^* = 0$ for $i = 1, 2$ we have (assuming $\gamma_b$ large enough): $\gamma_k \nabla f_{i_k}(x) = \frac{x-1}{2c_k}$, if $i_k = 1$, and $\gamma_k \nabla f_{i_k}(x) = \frac{x+1}{2c_k}$ if $i_k = 2$.

Crucially, note that this update is *curvature-independent*. The expected update is $\mathbb{E}_{i_k}[\gamma_k \nabla f_{i_k}(x)] = \frac{x-1}{4c_k} + \frac{x+1}{4c_k} = \frac{1}{2c_k}x$. Hence, the iterates can only converge to $x = 0$ — because this is the only fixed point for the update rule. The proof naturally extends to the multidimensional setting, an illustration can be found in Fig. 2.

In the same picture, we show how our modified variant of the vanilla stepsize — we call this new algorithm DecSPS, see §5 — instead converges to the correct solution.

*Remark* 2. SGD with (non-adaptive) stepsize $\gamma_k$ instead keeps the curvature, and therefore is able to correctly estimate the average $\mathbb{E}_{i_k}[\gamma_k \nabla f_{i_k}(x)] = \frac{\gamma_k}{2}(a_1 + a_2)\left[x - \frac{a_1-a_2}{a_1+a_2}\right]$ — precisely because $\gamma_k$ is independent from $\nabla f_{i_k}$. From this we can see that SGD can only converge to the correct stationary point $x^* = \frac{a_1-a_2}{a_1+a_2}$ — because again this is the only fixed point for the update rule.

In the appendix, we go one step further and provide an analysis of the bias of SPS in the one-dimensional quadratic case (Prop. 4). Yet, we expect the precise characterization of the bias phenomenon in the non-quadratic setting to be particularly challenging. We provide additional insights in §D.2. Instead, in the next section, we show how to effectively modify $\gamma_k$ to yield convergence to $x^*$ without further assumptions.

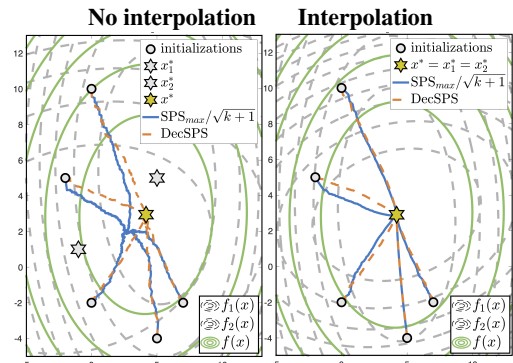

Figure 2: Dynamics of $\text{SPS}_{\max}$ with decreasing multiplicative constant (SGD style) compared with DecSPS. We compared both in the **interpolated setting (right)** and in the **non-interpolated setting (left)**. In the non-interpolated setting, a simple multiplicative factor introduces a bias in the final solution, as discussed in this section. We consider two dimensional $f_i = \frac{1}{2}(x - x_i^*)^\top H_i(x - x_i^*)$, for $i = 1, 2$ and plot the contour lines of the corresponding landscapes, as well as the average landscape $(f_1 + f_2)/2$ we seek to minimize. Solution is denoted with a gold star.

## 5 DecSPS: Convergence to the exact solution

We propose the following modification of the vanilla SPS proposed in [22], designed to yield convergence to the exact minimizer while keeping the main adaptiveness properties[4]. We call it Decreasing SPS (DecSPS), since it combines a steady stepsize decrease with the adaptiveness of SPS.

---

[4] Similar choices are possible. We found that this leads to the biggest stepsize magnitude, allowing for faster convergence in practice.

$$\gamma_k := \frac{1}{c_k} \min \left\{ \frac{f_{\mathcal{S}_k}(x^k) - \ell^*_{\mathcal{S}_k}}{\|\nabla f_{\mathcal{S}_k}(x^k)\|^2}, \ c_{k-1}\gamma_{k-1} \right\}, \qquad \text{(DecSPS)}$$

for $k \in \mathbb{N}$, where $c_k \neq 0$ for every $k \in \mathbb{N}$. We set $c_{-1} = c_0$ and $\gamma_{-1} = \gamma_b > 0$ (stepsize bound, similar to [22]), to get $\gamma_0 := \frac{1}{c_0} \cdot \min \left\{ \frac{f_{\mathcal{S}_0}(x^0) - \ell^*_{\mathcal{S}_0}}{\|\nabla f_{\mathcal{S}_0}(x^k)\|^2}, \ c_0\gamma_b \right\}$.

**Lemma 1.** *Let each $f_i$ be $L_i$ smooth and let $(c_k)_{k=0}^\infty$ be any non-decreasing positive sequence of real numbers. Under DecSPS, we have* $\min \left\{ \frac{1}{2c_k L_{\max}}, \frac{c_0 \gamma_b}{c_k} \right\} \leq \gamma_k \leq \frac{c_0 \gamma_b}{c_k}$, *and $\gamma_{k-1} \leq \gamma_k$*

*Remark* 3. As stated in the last lemma, under the assumption of $c_k$ non-decreasing, $\gamma_k$ is trivially *non-increasing* since $\gamma_k \leq c_{k-1}\gamma_{k-1}/c_k$.

The proof can be found in the appendix, and is based on a simple induction argument.

### 5.1 Convergence under bounded iterates

The following result provides a proof of convergence of SGD for the $\gamma_k$ sequence defined above.

**Theorem 3.** *Consider SGD with DecSPS and let $(c_k)_{k=0}^\infty$ be any non-decreasing sequence such that $c_k \geq 1, \forall k \in \mathbb{N}$. Assume that each $f_i$ is convex and $L_i$ smooth. We have:*

$$\mathbb{E}[f(\bar{x}^K) - f(x^*)] \leq \frac{2c_{K-1}\tilde{L}D^2}{K} + \frac{1}{K}\sum_{k=0}^{K-1} \frac{\hat{\sigma}_B^2}{c_k}, \qquad (4)$$

*where $D^2 := \max_{k \in [K-1]} \|x^k - x^*\|^2$, $\tilde{L} := \max \left\{ \max_i\{L_i\}, \frac{1}{2c_0\gamma_b} \right\}$ and $\bar{x}^K = \frac{1}{K}\sum_{k=0}^{K-1} x^k$.*

If $\hat{\sigma}_B^2 = 0$, then $c_k = 1$ for all $k \in \mathbb{N}$ leads to a rate $\mathcal{O}(\frac{1}{K})$, well known from [22]. If $\hat{\sigma}_B^2 > 0$, as for the standard SGD analysis under decreasing stepsizes, the choice $c_k = \mathcal{O}(\sqrt{k})$ leads to an optimal asymptotic trade-off between the deterministic and the stochastic terms, hence to the asymptotic rate $\mathcal{O}(1/\sqrt{k})$ since $\sum_{k=0}^{K-1} \frac{1}{\sqrt{k+1}} \leq 2\sqrt{K}$. Moreover, picking $c_0 = 1$ minimizes the speed of convergence for the deterministic factor. Under the assumption that $\hat{\sigma}_B^2 \ll \tilde{L}D^2$ (e.g. reasonable distance initialization-solution and $L_{\max} > 1/\gamma_b$), this factor is dominant compared to the factor involving $\hat{\sigma}_B^2$. For this setting, the rate simplifies as follows.

**Corollary 2.** *Under the setting of Thm. 3, for $c_k = \sqrt{k+1}$ ($c_{-1} = c_0$) we have*

$$\mathbb{E}[f(\bar{x}^K) - f(x^*)] \leq \frac{2\tilde{L}D^2 + 2\hat{\sigma}_B^2}{\sqrt{K}}. \qquad (5)$$

*Remark* 4 (Beyond bounded iterates). The result above crucially relies on the bounded iterates assumption: $D^2 < \infty$. To the best of our knowledge, if no further regularity is assumed, modern convergence results for adaptive methods (e.g. variants of AdaGrad) in convex stochastic programming require[5] this assumption, or else require gradients to be globally bounded. To mention a few: [11, 26, 32, 8, 31]. A simple algorithmic fix to this problem is adding a cheap projection step onto a large bounded domain [21]. We can of course include this projection step in DecSPS, and the theorem above will hold with no further modification. Yet we found this to be not necessary: the strong guarantees of SPS in the strongly convex setting [22] let us go one step beyond: in §5.2 we show that, if each $f_i$ is strongly convex (e.g. regularizer is added), then one can bound the iterates globally with probability one, without knowledge of the gradient Lipschitz constant. To the best of our knowledge, no such result exist for AdaGrad — except [30], for the deterministic case.

*Remark* 5 (Dependency on the problem dimension). In standard results for AdaGrad, a dependency on the problem dimension often appears (e.g. Thm. 1 in [31]). This dependency follows from a bound on the AdaGrad preconditioner that can be found e.g. in Thm. 4 in [21]. In the SPS case no such dependency appears — specifically because the stepsize is lower bounded by $1/(2c_k L_{\max})$.

---

[5] Perhaps the only exception is the result of [33], where the authors work on a different setting: i.e. they introduce the RUIG inequality.

## 5.2 Removing the bounded iterates assumption

We prove that under DecSPS the iterates live in a set of diameter $D_{\max}$ almost surely. This can be done by assuming strong convexity of each $f_i$.

The result uses this alternative definition of *neighborhood*: $\hat{\sigma}^2_{B,\max} := \max_{\mathcal{S}\subseteq[n],|\mathcal{S}|=B}[f_{\mathcal{S}}(x^*)-\ell^*_{\mathcal{S}}]$.

Note that trivially $\hat{\sigma}^2_{B,\max} < \infty$ under the assumption that all $f_i$ are lower bounded and $n < \infty$.

**Proposition 1.** *Let each $f_i$ be $\mu_i$-strongly convex and $L_i$-smooth. The iterates of SGD with DecSPS with $c_k = \sqrt{k+1}$ (and $c_{-1} = c_0$) are such that $\|x^k - x^*\|^2 \leq D^2_{\max}$ almost surely $\forall k \in \mathbb{N}$, where $D^2_{\max} := \max\left\{ \|x^0 - x^*\|^2, \frac{2c_0\gamma_b\hat{\sigma}^2_{B,\max}}{\min\left\{\frac{\mu_{\min}}{2L_{\max}},\mu_{\min}\gamma_b\right\}} \right\}$, with $\mu_{\min} = \min_{i\in[n]} \mu_i$ and $L_{\max} = \max_{i\in[n]} L_i$.*

The proof relies on the variations of constants formula and an induction argument — it is provided in the appendix. We are now ready to state the main theorem for the unconstrained setting, which follows from Prop. 1 and Thm. 3.

**Theorem 4.** *Consider SGD with the DecSPS stepsize $\gamma_k := \frac{1}{\sqrt{k+1}} \cdot \min\left\{ \frac{f_{\mathcal{S}_k}(x^k)-f^*_{\mathcal{S}_k}}{\|\nabla f_{\mathcal{S}_k}(x^k)\|^2}, \gamma_{k-1}\sqrt{k}\right\}$, for $k \geq 1$ and $\gamma_0$ defined as at the beginning of this section. Let each $f_i$ be $\mu_i$-strongly convex and $L_i$-smooth:*

$$\mathbb{E}[f(\bar{x}^K) - f(x^*)] \leq \frac{2\tilde{L}D^2_{\max} + 2\hat{\sigma}^2_B}{\sqrt{K}}. \tag{6}$$

*Remark* 6 (Strong Convexity). The careful reader might notice that, while we assumed strong convexity, our rate is slower than the optimal $\mathcal{O}(1/K)$. This is due to the adaptive nature of DecSPS. It is indeed notoriously hard to achieve a convergence rate of $\mathcal{O}(1/K)$ for adaptive methods in the strongly convex regime. While further investigations will shed light on this interesting problem, we note that *the result we provide is somewhat unique in the literature*: we are not aware of any adaptive method that enjoys a similar convergence rate without either (a) assuming bounded iterates/gradients or (b) assuming knowledge of the gradient Lipschitz constant or the strong convexity constant.

*Remark* 7 (Comparison with Vanilla SGD). On a convex problem, the non-asymptotic performance of SGD with a decreasing stepsize $\gamma_k = \eta/\sqrt{k}$ strongly depends on the choice of $\eta$. The optimizer might diverge if $\eta$ is too big for the problem at hand. Indeed, most bounds for SGD, under no access to the gradient Lipschitz constant, display a dependency on the size of the domain and rely on projections after each step. If one applies the method in the unconstrained setting, such convergence rates technically do not hold, and tuning is sometimes necessary to retrieve stability and good performance. Instead, for DecSPS, simply by adding a small regularizer, the method is guaranteed to converge at the non-asymptotic rate we derived even in the unconstrained setting.

## 5.3 Extension to the non-smooth setting

For any $\mathcal{S} \subseteq [n]$, we denote in this section by $g_{\mathcal{S}}(x)$ the subgradient of $f_{\mathcal{S}}$ evaluated at $x$. We discuss the extension of DecSPS to the non-smooth setting.

A straightforward application of DecSPS leads to a stepsize $\gamma_k$ which is no longer lower bounded (see Lemma 1) by the positive quantity $\min\left\{ \frac{1}{2c_k L_{\max}}, \frac{c_0\gamma_b}{c_k}\right\}$. Indeed, the gradient Lipschitz constant in the non-smooth case is formally $L_{\max} = \infty$. Hence, $\gamma_k$ prescribed by DecSPS can get arbitrarily small[6] for finite $k$. One easy solution to the problem is to enforce a lower bound, and adopt a new proof technique. Specifically we propose the following:

$$\gamma_k := \frac{1}{c_k} \cdot \min\left\{ \max\left\{ c_0\gamma_\ell, \frac{f_{\mathcal{S}_k}(x^k)-\ell^*_{\mathcal{S}_k}}{\|g_{\mathcal{S}_k}(x^k)\|^2}\right\}, c_{k-1}\gamma_{k-1}\right\}, \qquad \text{(DecSPS-NS)}$$

---

[6] Take for instance the deterministic setting one-dimensional setting $f(x) = |x|$. As $x \to 0$, the stepsize prescribed by DecSPS converges to zero. This is not the case e.g. in the quadratic setting.

where $c_k \neq 0$ for every $k \geq 0$, $\gamma_\ell \geq \gamma_b$ is a small positive number and all the other quantities are defined as in DecSPS. In particular, as for DecSPS, we set $c_{-1} = c_0$ and $\gamma_{-1} = \gamma_b$. Intuitively, $\gamma_k$ is selected to live in the interval $[c_0\gamma_\ell/c_k, c_0\gamma_b/c_k]$ (see proof in §F, appendix), but has subgradient-dependent adaptive value. In addition, this stepsize is enforced to be monotonically decreasing.

**Theorem 5.** *For any non-decreasing positive sequence $(c_k)_{k=0}^\infty$, consider SGD with DecSPS-NS. Assume that each $f_i$ is convex and lower bounded. We have*

$$\mathbb{E}[f(\bar{x}^K) - f(x^*)] \leq \frac{c_{K-1}D^2}{\gamma_\ell c_0 K} + \frac{1}{K}\sum_{k=0}^{K-1}\frac{c_0\gamma_b G^2}{c_k}, \tag{7}$$

*where $D^2 := \max_{k\in[K-1]}\|x^k - x^*\|^2$ and $G^2 := \max_{k\in[K-1]}\|g_{\mathcal{S}_k}(x^k)\|^2$.*

One can then easily derive an $\mathcal{O}(1/\sqrt{k})$ convergence rate. This is presented in §F (appendix).

## 6 Numerical Evaluation

We evaluate the performance of DecSPS with $c_k = c_0\sqrt{k+1}$ on binary classification tasks, with regularized logistic loss $f(x) = \frac{1}{n}\sum_{i=1}^n \log(1 + \exp(y_i \cdot a_i^\top x)) + \frac{\lambda}{2}\|x\|^2$, where $a_i \in \mathbb{R}^d$ is the feature vector for the $i$-th datapoint and $y_i \in \{-1, 1\}$ is the corresponding binary target.

We study performance on three datasets: (1) a Synthetic Dataset, (2) The A1A dataset [6] and (3) the Breast Cancer dataset [10]. We choose different regularization levels and batch sizes bigger than 1. Details are reported in §G, and the code is available at https://github.com/aorvieto/DecSPS. At the batch sizes and regularizer levels we choose, the problems do not satisfy interpolation. Indeed, running full batch gradient descent yields $f^* > 0$. While running $\text{SPS}_{\max}$ on these problems (1) does not guarantee convergence to $f^*$ and (2) requires full knowledge of the set of optimal function values $\{f_{\mathcal{S}}^*\}_{|\mathcal{S}|=B}$, in DecSPS we can simply pick the lower bound $0 = \ell_{\mathcal{S}}^* \leq f_{\mathcal{S}}^*$ for every $\mathcal{S}$. Supported by Theorems 3 & 4 & 5, we expect SGD with DecSPS to converge to the minimum $f^*$.

**Stability of DecSPS.** DecSPS has two hyperparameters: the upper bound $\gamma_b$ on the first stepsize and the scaling constant $c_0$. While Thm. 5 guarantees convergence for any positive value of these hyperparameters, the result of Thm. 3 suggests that using $c_0 = 1$ yields the best performance under the assumption that $\hat{\sigma}_B^2 \ll \tilde{L}D^2$ (e.g. reasonable distance of initialization from the solution, and $L_{\max} > 1/\gamma_b$). In Fig. 3, we show on the synthetic dataset that (1) $c_0 = 1$ is indeed the best choice in this setting and (2) the performance of SGD with DecSPS is almost independent of $\gamma_b$. Similar findings are reported and commented in Figure 7 (Appendix) for the other datasets. Hence, *for all further experiments*, we choose the hyperparameters $\gamma_b = 10, c_0 = 1$.

**Comparison with vanilla SGD with decreasing stepsize.** We compare the performance of DecSPS against the classical decreasing SGD stepsize $\eta/\sqrt{k+1}$, which guarantees convergence to the exact solution at the same asymptotic rate as DecSPS. We show that, while the asymptotics are the same, DecSPS with hyperparameters $c_0 = 1, \gamma_b = 10$ performs competitively to a fine-tuned $\eta$ — where crucially the optimal value of $\eta$ depends on the problem. This behavior is shown on all the considered datasets, and is reported in Figures 4 (*Breast* and *Synthetic* reported in the appendix for space constraints). If lower regularization ($1e-4, 1e-6$) is

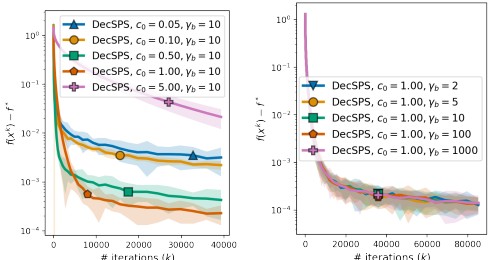

Figure 3: DecSPS ($c_k = c_0\sqrt{k+1}$) sensitivity to hyperparameters on the *Synthetic Dataset*, with $\lambda = 0$. Repeated 10 times and plotted is mean and std.

considered, then DecSPS can still match the performance of tuned SGD — but further tuning is needed (see Figure 14. Specifically, since the non-regularized problems do not have strong curvature, we found that DecSPS works best with a much higher $\gamma_b$ parameter and $c_0 = 0.05$.

**DecSPS yields a truly adaptive stepsize.** We inspect the value of $\gamma_k$ returned by DecSPS, shown in Figures 4 & 8 (in the appendix). Compared to the vanilla SGD stepsize $\eta/\sqrt{k+1}$, a crucial difference appears: $\gamma_k$ decreases faster than $\mathcal{O}(1/\sqrt{k})$. This showcases that, while the factor $\sqrt{k+1}$

can be found in the formula of DecSPS[7], the algorithm structure provides additional adaptation to curvature. Indeed, in (regularized) logistic regression, the local gradient Lipschitz constant increases as we approach the solution. Since the optimal stepsize for steadily-decreasing SGD is $1/(L\sqrt{k+1})$, where $L$ is the global Lipschitz constant [13], it is pretty clear that $\eta$ should be decreased over training for optimal converge (as $L$ effectively increases). This is precisely what DecSPS is doing.

**Comparison with AdaGrad stepsizes.** Last, we compare DecSPS with another adaptive coordinate-independent stepsize with strong theoretical guarantees: the norm version of AdaGrad (a.k.a. AdaGrad-Norm, AdaNorm), which guarantees the exact solution at the same asymptotic rate as DecSPS [32]. AdaGrad-norm at each iteration updates the scalar $b_{k+1}^2 = b_k^2 + \|\nabla f_{\mathcal{S}_k}(x_k)\|^2$ and then selects the next step as $x_{k+1} = x_k - \frac{\eta}{b_{k+1}}\nabla f_i(x_k)$. Hence, it has tuning parameters $b_0$ and $\eta$. In Fig. 4 we show that, on the Breast Cancer dataset, after fixing $b_0 = 0.1$ as recommended in [32] (see their Figure 3), tuning $\eta$ cannot quite match the performance of DecSPS. This behavior is also observed on the other two datasets we consider (see Fig. 9 in the Appendix). Last, in Fig. 10 & 11 in the Appendix, we show that further tuning of $b_0$ likely does not yield a substantial improvement.

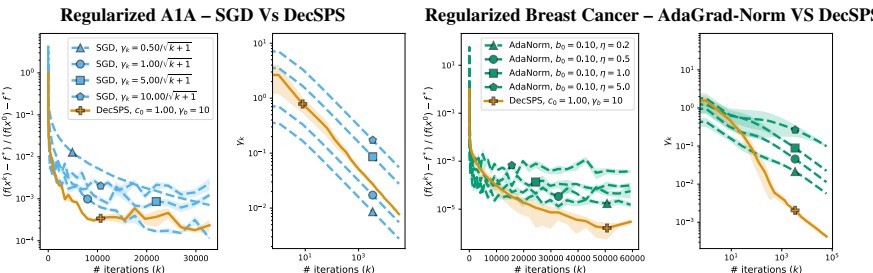

Figure 4: **Left:** performance of DecSPS, on the A1A Dataset ($\lambda = 0.01$). **Right:** performance of DecSPS on the Breast Cancer Dataset ($\lambda = 1e-1$). Further experiments can be found in §G (appendix).

**Comparison with Adam and AMSgrad without momentum.** In Figures 5 & 12 & 13 we compare DecSPS with Adam [19] and AMSgrad [26] on the A1A and Breast Cancer datasets. Results show that DecSPS with the usual hyperparameters is comparable to the fine-tuned version of both these algorithms — which however do not enjoy convergence guarantees in the unbounded domain setting.

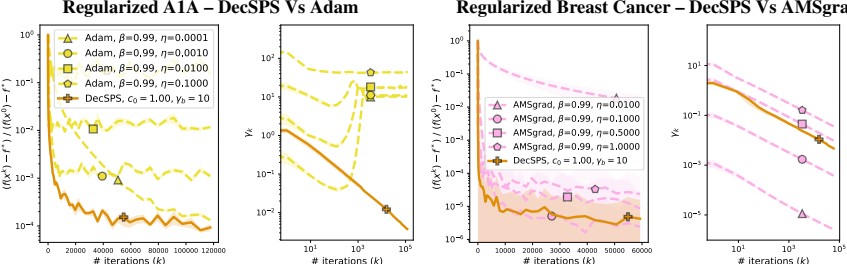

Figure 5: **Left**: Performance of Adam (with fixed stepsize and no momentum) and **Right**: AMSgrad (with sqrt decreasing stepsize and no momentum) compared to DecSPS on the A1A and Breast Cancer dataset, respectively. Plots comparing the performance of Adam with DecSPS on the Breast Cancer Dataset can be found in Figure 13, and plots comparing AMSgrad with DecSPS on the A1A Dataset can be found in Figure 12. Plotted is also the average stepsize (each parameter evolves with a different stepsize).

## 7 Conclusions and Future Work

We provided a practical variant of SPS [22], which converges to the true problem solution without the interpolation assumption in convex stochastic problems — matching the rate of AdaGrad. If in addition, strong convexity is assumed, then we show how, in contrast to current results for AdaGrad, the bounded iterates assumption can be dropped. The main open direction is a proof of a faster rate $\mathcal{O}(1/K)$ under strong convexity. Other possible extensions of our work include using the proposed new variants of SPS with accelerated methods, studying further the effect of mini-batching and non-uniform sampling of DecSPS, and extensions to the distributed and decentralized settings.

---

[7] We pick $c_k = c_0\sqrt{k+1}$, as suggested by Cor. 2 & 3

## Acknowledgements

This work was partially supported by the Canada CIFAR AI Chair Program. Simon Lacoste-Julien is a CIFAR Associate Fellow in the Learning in Machines & Brains program.

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
