# OpenReview forum: "Dynamics of SGD with Stochastic Polyak Stepsizes: Truly Adaptive Variants and Convergence to Exact Solution"
_NeurIPS.cc/2022/Conference — NeurIPS 2022 Accept_

### Official Review · Reviewer_bV6Q · 2022-07-01

**Rating:** 6
**Confidence:** 4
**Soundness:** 3 good
**Presentation:** 3 good
**Contribution:** 2 fair

**Summary:**


This paper proposes DecSPS, which aims at improving the convergence of SPS in two aspects: (1) removing the optimal minibatch loss in the step sizes by replacing it with a lower bound, and (2) proving that DecSPS converges to the exact minimizer in the strongly convex setting by showing that the iterates are bounded in the strongly convex setting.

**Questions:**


Please see above.

**Limitations:**

Yes

**Strengths And Weaknesses:**


Strengths:

1. The overall writing is clear and easily understandable, and the proof is also well-organized.
2. The contributions of this work are also clearly stated.

Major concerns:

1. Overall, the theoretical contributions are not very significant. The proof of the almost sure boundedness of iterates (Proposition 5.4) is novel, which helps establish that DecSPS can reach the exact minimizer under strong convexity. On the other hand, removing the optimal minibatch loss in the step sizes by replacing it with a lower bound is nice in practice, but does not need a complicated adjustment in the proof.

More results can be added to this work if possible. For example, does DecSPS have similar properties as in Theorem 3.1 and 3.8 of [22] for strongly convex and nonconvex settings?

2. In the numerical results, comparisons with other adaptive methods such as Adam and RMSProp should also be added.
3. Also, more numerical results on benchmarks of deep learning can be added, e.g. ResNet56 on Cifar10, which nearly satisfies the interpolation condition. Does DecSPS have an advantage over other adaptive methods?

Minor concerns:

1. It would be better if Table 1 can be put in the main text, as it provides a succinct summary of the main results.
2. The role of parameter $\gamma_b$ in SPS and DecSPS should be explained. This parameter follows from [22], but it should still be introduced and discussed here.
3. The fonts of legends and x, y axis should be larger in the figures.
4. In line 227, the $k$ in the formula should be $0$
5. In line 282, and or should be and/or

---

> ### Author Response · Authors · 2022-08-02
> **Thank you for your comments and suggestions**
>
> We thank the reviewer for the suggestions on our paper. You can find our replies below:
> 1) **Significance of theoretical contributions**: Some (not all) of our contributions have an associated easy proof. Yet, we politely disagree with the reviewer in that a piece of work is valuable only if it has complex proof. Replacing the $f_i*$ with a lower bound is a simple modification of the proof of Loizou et al. 2021. Yet, it was not noted in their paper and would have made their contribution undoubtedly stronger. Further, we note that both Theorem 5.2 and Proposition 5.4 are novel proof-wise: merge an Adagrad-like argument with some SPS properties – to the best of our knowledge, no paper combined these techniques in the literature. Also, the discussion on the non-convergence of SPS-max in the non-interpolated setting is novel and, we think, particularly stimulating. Again, we do not believe a result has to necessarily have pages and pages of complex proof to be valuable. Nonetheless, we believe our result stands out in the literature since DecSPS is the first stochastic adaptive optimization method that converges to the exact solution without restrictive assumptions like bounded iterates/gradients.
> 2) **Convergence in non-convex and strongly convex setting**: As discussed in the conclusion section, $O(1/k)$ convergence in the strongly-convex setting remains an open problem. A result in this setting for sure requires new proof techniques since results in the strongly-convex setting are not developed in the adaptive methods literature (e.g., we are not aware of any such result for Adam or RMSprop). Regarding the non-convex case, this is not a setting we discuss in this paper, since most non-convex problems of interest like training a neural network do satisfy interpolation.  We also note that in the original paper of Loizou et al. this scenario has strong theoretical limitations (knowledge of the gradient Lipschitz constant is required).
> 3) **Comparison with Adam**: *We already compared against Adam and AMSgrad in the appendix (see figures 10-11)*
> 4) **Deep learning experiments**: Deep learning problems are most times over-parametrized hence they satisfy interpolation. Hence, one should use the original SPS_max by Loizou et al. (2021) for these problems. Note also that schedulers often studied in optimization to derive rates, such as a steady $O(1/\sqrt k)$ decrease, are often too drastic in deep learning settings, where the stepsize is only reduced after a sizeable number of epochs (see e.g., the common cosine decay scheduler).
> 5) **Minor concerns**: Thank you a lot for these points; we will implement all of them in a potential camera-ready version.
>
> If you agree that we managed to address all issues, please consider raising your mark. If you believe this is not the case, please let us know so that we have a chance to respond.

---

> > ### Comment · Reviewer_bV6Q · 2022-08-07
> > **Thanks for your response**
> >
> >
> > Sorry for missing the comparisons with Adam and AMSgrad in the paper!
> >
> > I do agree that significant results do not necessarily come with complicated proofs. However, the proof of replacing $f_i^*$ with a lower bound still looks almost the same as the original proof of (Loizou et al.). The fact that we can use a lower bound is nice in practice but the theoretical result is not very surprising to me. I have acknowledged in my review that the proof of Proposition 5.4 is new.
> >
> > For "Convergence in strongly convex setting", what I wanted to ask is: can DecSPS get a result as in Theorem 3.1 of (Loizou et al.), that is, linear convergence to a neighborhood of the solution, but not the $O(1/\sqrt(K))$ rate. I agree that in the nonconvex case, (Loizou et al.) needs to know the gradient Lipschitz constant.
> >
> > I have raised my score to 6.

---

> > > ### Author Response · Authors · 2022-08-07
> > > **Thank you for engaging**
> > >
> > > Thank you so much for engaging in the discussion and raising your score!
> > >
> > > - **Regarding** $f_i^* \to l_i^*$ : you are right in saying that replacing this in the proofs of Loizou et. al is easy. However, we found this fact to be surprising, and could not find a similar "trick" in the literature. We think this significantly improves the applicability of SPS, and it is particularly interesting in light of many other attempts that try to solve the "knowledge of $f_i^*$" problem, e.g. https://arxiv.org/pdf/1905.00313.pdf (Algorithm 3).
> > >
> > > - **Regarding the linear rate**: If $c_k$ is constant, then DecSPS reduces to SPS and, for strongly convex functions, we retrieve a linear rate to a ball around the solution (if we are not overparametrized). However, from the classical analysis of SGD we know that as soon as one reduces the stepsize overtime, the linear rate is lost --- even in the noiseless setting. Two classic papers on the subject are https://www.di.ens.fr/~fbach/gradsto_nips2011.pdf and https://arxiv.org/abs/1212.2002 . Therefore, if one cares about convergence to the exact solution and the problem is not overparametrized, then we would advise selecting an increasing $c_k$. Otherwise, we would recommend using a constant $c_k$ to get back to SPS_max by Loizou et al.
> > >
> > > Please do not hesitate to contact us for further questions, and thanks again!

---

### Official Review · Reviewer_CpM5 · 2022-07-04

**Rating:** 8
**Confidence:** 5
**Soundness:** 4 excellent
**Presentation:** 4 excellent
**Contribution:** 4 excellent

**Summary:**

This paper presents a novel variant of Stochastic Polyak Stepsizes (SPS) providing adaptation when the usual interpolation assumption does not hold, as soon as a lower bound of the loss is known.

The authors explain first that traditional SPS methods
- (1) require the knowledge of the optimal loss on any batch which is often not the case and
- (2) only converge theoretically to a neighborhood of the solution

Authors provide a practical fix which is to replace the optimal loss on any batch by a lower bound of the loss (often 0 as many losses are non-negative) and call this algorithm SPS_max^l as it is a variant of SPS_max proposed in [Loizou et al, 2021]. Moreover, to enforce convergence to the solution, they make the upper bound of the adaptive step size decrease at each iteration in O(1/sqrt(k)) and provide convergence guarantees in O(1/sqrt(K)) of this new algorithm call DecSPS:
- for convex and smooth losses with a lower bound and bounded iterates
- for strongly convex and smooth losses with a lower bound (thus removing the bounded iterate assumption)
Finally, they extend their method to the non-smooth setting by enforcing a lower bound on their SPS version leading to DecSPS-NS and prove also O(1/sqrt(K)) convergence for
- convex non-smooth losses with a lower bound, bounded iterates and bounded subgradients

Finally, authors run experiments on regularized logistic regression on 3 datasets
- one small synthetic
- two small real datasets A1A and Breast Cancer
 for which interpolation is not met. They compare the performance of DecSPS with AdaGrad-Norm and SGD with decreasing

They also give elements indicating that their method is not too sensible to \gamma_b : the upper bound of DecSPS and that the decrease of the step size thus computed is faster than O(1/sqrt(k)) which shows real curvature adaptation.

~~~~
Score updated to 8 to support the paper after the answer of the authors to the question "What would be the downsides of using DecSPS or SPS instead of SGD?" (by reviewer 98CJ).

**Questions:**

The paper is very clear, reasoning is easy to follow and going though the proofs was a pleasure.

Let me start with typos and nitpicking:
- A1) Only cited equations should be numbered. Useful cited equations are flooded in all the numbered equations (makes it harder to navigate in the proofs)
- A2) line 68 : "Dependency on f_{S_k}^*" star is missing
- A3) lines 102-105 : precise "in section 4"
- A4) line 167 & 172 : strange too read "$f_i$s"
- A5) line 171 : "linear rate" right ?
- A6) line 229: gamma is decreasing not increasing (this typo appears several times eg in E.1)
- A7) line 232: "appendix E.1" to simplify the reading
- A8) line 240: Strange way of calling both terms in (4) : both are deterministic. Better to say "distance-to-the-solution term"  and a "noise" term
- A9) line 243 : strange "distance initialization-solution" -> instead initialization-solution distance ?
- A10) line 254 : "regularizer is added" and then interpolation is thus lost, right ?
- A11) line 273 : better to stick with notations with c_k and c_{k-1}. There is no point changing this only here
- A12) Footnote 3 page 7 : miscitation. [31] should be cited here. No RUIG inequality in [30]
- A13) In Lemma E.1, I do not think $a \leq 1$ is required. If it were, then it would imply that one would need to set $\gamma_b$ such that $\mu_{\min} \gamma_b \leq 1$ in (114)
- A14) Eq 107 Minus sign missing and I would put a $\leq$ at the RHS
- A15) Same typo : step sizes are decreasing not increasing
- A16) line 762 : \alpha_k are non-negative
- A17) eq (126) I think a constant 2 is lost on the left-hand side. It would give a sharper result, nothing important asymptotically
- A18) line 783 : J*E*nsen not Jansen


More major questions :
- B1) I am not sure to be aware of all results in adaptive and SPS methods so I wonder what are the known classical results in the domain. A small table like in A.1 would have helped summarizing assumptions and known results for methods cited in the second paragraph of 1.1. I hope the other reviewers will have a clear opinion whether this convergence result without bounded iterates nor gradients nor using the smoothness constant nor the strong convexity value is novel or not. If not, it would worth citing relevant works.
- B2) line 325 : Could you precise why you say that \gamma_b = 10 is "theoretically inspired" ? I find it a bit incorrect.
- B3) line 334-335 : Could you provide a reference or a quick explanation why the local gradient Lipshchitz constant increases near the solution ?
- B4) As pointed out in weaknesses, the regularization parameters used are way too large. The problems become very simple too solve and as we are talking about optimization methods for Machine Learning than we should keep in mind that our goal is to have good generalization of our model and I doubt that this is the case here. I would like to see the results for \lambda = 1e-4 and 1e-6.
- B5) Lack of consistency in the setting of hyperparameters depending on the methods: DecSPS parameters are set on the real datasets A1A and Breast whereas the b_0 for Adagrad-norm is set on the synthtetic data set... Why ?


**Limitations:**

We would definitely need to see how pratical this new methods are in practice on large models or at least on difficult logistic regression problems (easy to find real-sim or rcv1.binary dataset on LIBSVM Data: Classification (Binary Class) website)

I already mentioned this in weaknesses but the DecSPS method comes with 2 or 3 hyper-parameters to tune which I think is not in fact taking us away from "adaptivity".

Not really a limitation but potential extension: as the authors point out in their conclusion:
- they still unable to prove O (1/K) convergence for strongly convex objective

**Strengths And Weaknesses:**

Strengths:
- Problem and interest of the paper well posed
- Good review of existing methods (in the appendix for more details) in the interpolation case
- Solid theoretical guarantees (I went though all the proofs and everything looks correct)
- The paper is particularly well written, good intuition is given in the proofs which are easy to follow and makes the intentions clear to the reader


Weaknesses:
- The experimental section is very light. I know it's a common (and sometimes easy criticism) but I was a bit disappointed by the experiments which are focusing on regularized logistic regression on 3 small datasets. See experiments on CIFAR-10 or CIFAR-100 of [1], [2] and [3]
- the regularization in the experiments is way too large in my opinion. Makes the problems solved way too simple...
- A paper proposing a new SPS methods without comparing it *numerically* to previous variants (SPS_max [1], Ali-G [2], SPSL1 from [1] and maybe other) is a bit problematic. I would have expected experiments on a "gradient" of experiments from non interpolated to interpolated like done in the motivational plot (but on a larger problem) by playing on the regularization.
- It looks like the price to pay for "adaptivity" is to tune 2 parameters (c_0 and \gamma_b) or 3 (additional \gamma_l) in the non-smooth case. The authors try to show numerically that numerical results are not too sensitive to these values but as in previous points, Imo experiments are not exhaustive enough to conclude.

[1] Loizou et al, Stochastic Polyak Step-size for SGD: An Adaptive Learning Rate for Fast Convergence (2021)
[2] Berrada et al, Training Neural Networks for and by Interpolation (2020)
[3] Gower et al, Cutting Some Slack for SGD with Adaptive Polyak Stepsizes (2022)

---

> ### Author Response · Authors · 2022-08-02
> **Thank you for your very thorough review!**
>
> We thank the reviewer for the nice summary of our paper and the many constructive comments. We are glad you found our paper interesting.
> You can find an answer to all of your concerns below:
>
> 1) **More Experiments**: Our experiments align with the setting studied in the paper: convergence to the exact solution in stochastic non-interpolated convex optimization. Since methods like SPS_max, Ali-G and SPSL do not guarantee convergence to the exact solution (we provided an example in the bias section for SPS),  in our further comparison we did not include them. Instead, we compare with methods with convergence guarantees in this setting: SGD with decreasing stepsize, Adagrad, and AMSgrad (in the appendix). We think our experimental section is not light since we do compare on 3 problems against tuned versions of four other algorithms (see all plots in the appendix). We find that it is rare to see such detailed comparisons in the literature. Regarding neural networks: here, convergence to the exact solution is not crucial for performance, or interpolation is satisfied: hence, standard SPS would probably perform better. Similarly, one should not use SGD with stepsize 1/sqrt(k) in deep learning — different, less drastic, schedules are used: cosine decay, reduce_on_plateau, one-cycle, etc.. We remark that here we focus on the non-interpolated setting, and we will add a remark stating that in the non-interpolated settings, standard SPS_max is preferable.
> 2) **Experiments on other datasets**: given the time constraints we, unfortunately, cannot set up in a few days experiments on other datasets and show you the results now. But we will, in the potential camera ready, include the results for another dataset in LIBSVM.
> 3) **Performance under less regularization**: If a vanishing lambda is used, then it is possible for the problems at hand to get a very solid performance with SGD starting from a very big stepsize (like 100) and decreasing like 1/sqrt(k). This stepsize results in a non-monotonically decreasing loss, as can be seen e.g., in Figure 1 in the classical paper https://arxiv.org/abs/1212.2002 . If one equips DecSPS with a larger gamma_b and a smaller c_0, then the performance of SGD with decreasing stepsize can be matched. However, since the problem in this case almost satisfies interpolation, we would suggest using standard SPS --- that is why we selected a large lambda in the first place, to avoid interpolation. We provided an additional figure after the checklist of our updated version. It is of the best interest for us to provide the most solid comparisons, and your question is definitely interesting and worth answering with additional experiments, which we will include in a potential camera-ready version.
> 4) **Typos**: Thank you, we will implement all your comments in the potential camera ready! Here we reply to the major points and simply correct the other typos: A1) We agree and will modify this! A5) yes. A6) You are right; sorry for the confusion A8-A9) Nice suggestions, thanks! A10) Yes, exactly! A11) Thank you for spotting this inconsistency. A12) You are right, thank you! A13) Yes, we also spotted that it is not needed a few days after the submission. Thank you for your attention! A14) Done now. A17) Right! Modified.
> 5) **B1) Table with works on adaptive methods**: If a table manages to fit in the main paper, we will very gladly include it. Thanks for the suggestion. Regarding novelty: we performed an extensive literature review while writing the paper and believe our method is the only adaptive algorithm that does not require a bounded domain for convergence in the convex setting.
> 6) **B2) Gamma_b theoretically motivated**: You are right; we are sorry for the confusion. We will include instead the following sentence: “The value for c_0 is theoretically inspired (line 314-315) and leads to fast convergence. In addition we selected gamma_b=10 to give the method a sizeable slack, since gamma_b bounds the step from above.”
> 7) **B3) “Why does the Grad Lip constant increase around solution?”**: This can be seen simply by taking the toy loss $\log(1+\exp(x)) + \lambda x^2/2$, which is similar mathematically to the logistic loss. Here, a simple second derivative computation shows that far from the solution the curvature is $\lambda$, while it is much bigger at the solution.
> 8) **B4) Results for small regularizers** : See point 2 above
> 9) **B5) b0 tuned on synthetic data?** At the end of the paper, after the checklist, you can now find the plots you requested: even on the other datasets tuning b_0 does not help Adagrad norm, and DecSPS still performs best.

---

> > ### Comment · Reviewer_CpM5 · 2022-08-08
> > **Thanks for the response**
> >
> > Thank you for your answers and explanations (B2, B3 & B5).
> >
> > I understand you point concerning comparison with other variants of SPS. I would recommend adding your sentence "Our experiments align with the setting studied in the paper: convergence to the exact solution in stochastic non-interpolated convex optimization. Since methods like SPS_max, Ali-G and SPSL do not guarantee convergence to the exact solution (we provided an example in the bias section for SPS), in our further comparison we did not include them" to the end of Section A.2.2 then.
> >
> > I still recommend this paper if few additional experiments are performed on a) more complex datasets and b) with smaller regularization.
> >
> > I won't raise my score for now but I hope other reviewers will when seeing your answers and my enthusiasm about the paper.

---

### Official Review · Reviewer_98CJ · 2022-07-13

**Rating:** 6
**Confidence:** 3
**Soundness:** 3 good
**Presentation:** 3 good
**Contribution:** 3 good

**Summary:**

This paper presents a method called DecSPS, which is based on a modification of stochastic Polyak stepsize (SPS). The proposed DecSPS method provides an exact convergence to the optimum value and is not dependent on prior knowledge of problem parameters. As a result, DecSPS achieves an exact solution without any restrictive assumptions such as bounded gradients and is the first stochastic adaptive optimization method.

**Questions:**

- How about the time complexity? Does it take more time than the existing algorithm?
- Should we directly start using SPS, or present the algorithm DecSPS instead of other methods like SGD with diminishing step sizes?



**Limitations:**

The limitations are briefly discussed within the paper, both in Introduction and conclusion. There is no negative societal impact of work.

**Strengths And Weaknesses:**

## STRENGTHS

- The modification performed on top SPS is minor and can be easily implemented.
- The paper is well written, structure is easy to follow.

## WEAKNESSES
- In figure 2, the example consists of 2 variables. It might be a good starting point but it is essential to have a model which can be adapted to larger scale experiments. Now we can run the experiments in a cloud system,..
- Minor typo :
... is bigger that the one for SPS_{\max}...
... is bigger **than** the one for SPS_{\max}...
- In Figure 3. sensitivity analysis could be performed for more seeds, 3 might be too less, it would be nice to see an error bar.
- The overall contribution is not too effective.

---

> ### Author Response · Authors · 2022-08-02
> **Reply to your questions**
>
> We thank the reviewer for the interesting questions and remarks, you can find our answer to your points below:
> 1) **“Figure 2 is only in 2 dimensions”**. The only purpose of Figure 2 is to illustrate the convergence issue of standard SPS with decreasing factors. In other words, it provides a counterexample for convergence and motivates our theory. Later, in the experiments section and appendix, we show the performance of our method on higher dimensional problems.
> 2) **Minor typo**: thanks! We will fix that.
> 3) **“Would be nice to see error bars in Figure 3”**: The performance on the synthetic dataset is quite stable as we change seeds. We provide a figure in the updated main paper at the end after the checklist: we did run the experiment 10 times and showed here 3 standard deviations. We will update the main-paper figure in the potential camera-ready.
> 4) **Time complexity**: In practice, our implementation runs in time comparable to SGD and Adagrad. Note that, in terms of number of gradient evaluations, the performance is exactly the same (B gradients per step, where B is the batch-size).. All other factors influencing the wall-clock time performance are negligible compared to the cost of computing the gradient (which requires matrix multiplications to compute predictions).
> 5) "**Should we directly start using SPS, or present the algorithm DecSPS instead of other methods like SGD with diminishing step sizes?**" We, unfortunately, do not fully understand this question, would you please rephrase it? DecSPS should be used in cases where we are interested in convergence to the exact solution, but quantities like smoothness parameter $L$ and stong convexity parameter $\mu$ are not available. In addition, when the interpolation condition is not satisfied, DecSPS should be preferred compare to SPS.

---

> > ### Comment · Reviewer_98CJ · 2022-08-08
> > **Reply to authors**
> >
> > Thank you for clearly pointing out the answers.
> >
> > "Should we directly start using SPS, or present the algorithm DecSPS instead of other methods like SGD with diminishing step sizes?"
> > To paraphrase my question:
> > What would be the downsides of using DecSPS or SPS instead of SGD? Do you foresee any problems or can we easily replace SGD with your method without any further restriction?
> >
> > Thanks.

---

> > > ### Author Response · Authors · 2022-08-08
> > > **Thanks for your question**
> > >
> > > Thank you so much for your reply and for rephrasing your question. It is an important one.
> > >
> > > First, it is clear that in convex stochastic non-interpolated problems, one has to decrease the stepsize over time for convergence to the exact solution --- it is a classical result from [A]. For SGD, e.g., [B] shows (but can be found in earlier references) that a decay rate of $O(1/\sqrt{k})$ is optimal among all schedules of the form $O(1/ k^\alpha)$, in the worst case. This leads to a rate of $O(1/\sqrt{k})$ in function value, in expectation.
> > > **Taking then SGD** with a decreasing stepsize $\eta/ \sqrt{k}$ (or $\eta/k$ if the problem is strongly convex [C]), the non-asymptotic performance ultimately depends on the choice of $\eta$. The optimizer might diverge if $\eta$ is too big for the problem at hand (usually unknown). That is why most bounds for SGD, _if one does not have access to the gradient Lipschitz constant_, have a dependency on the size of the domain and rely on projections after each gradient step. Perhaps the simplest bound can be found in [D], Equation 2.2.1. If stochastic gradients are bounded in squared norm by a constant $M^2$, and one projects the SGD update on a convex domain with size $D$ containing the solution, then SGD with stepsize $\frac{D}{M \sqrt{k}}$ converges as
> > >
> > > $$\mathbb{E}[f(\tilde x_k)]− f(x^*) \le \frac{D M}{\sqrt{k}}.$$
> > >
> > > If one applies the method in the unconstrained setting, the convergence rate _technically does not hold_ ($D=\infty$), and tuning is sometimes necessary to retrieve stability and good performance. Clearly, if $\eta<1/L$ ($L$ is the gradient Lip constant), the method will converge in expectation. However, one does not usually have access to $L$.
> > >
> > > **Instead, for DecSPS**, simply by adding a small regularizer, the method is guaranteed to converge at the $O(1/\sqrt{k})$ rate we derived even in the unconstrained setting --- no tuning of the stepsize is required in the worst case. This is not true for methods like AMSgrad and Adagrad, where $D$ or $M$ always appear in the bounds [E]. Therefore, in the worst case, these methods are not guaranteed to converge without a projection step.
> > >
> > > We are unaware of any adaptive method with such a strong convergence property as DecSPS. In addition, we showed in the experimental section that DecSPS can adapt to curvature in a non-trivial way.
> > >
> > > So, **in conclusion**, we are very excited about DecSPS and would advise using it in concrete applications where one needs a reliable optimizer with solid theoretical guarantees in the unconstrained setting. That said, **downsides**: we find it possible that fine-tuning SGD or Adagrad might have a slight edge for some problems in terms of speed. However, tuning is not possible in many settings: e.g., a new convex stochastic problem is to be solved reliably every few seconds as a subroutine on a more complex task.
> > >
> > > We hope that with this reply, we got you intrigued by the method, and we hope it will be helpful in your research as well.
> > >
> > > **If you also find it worthy of the NeurIPS publication**, we would kindly ask you to raise your score to increase our chances --- would be appreciated!
> > >
> > > Thank you, The Authors.
> > >
> > > [A] Kushner, Harold, and G. George Yin. Stochastic approximation and recursive algorithms and applications. Vol. 35. Springer Science & Business Media, 2003.
> > > [B] Moulines, Eric, and Francis Bach. "Non-asymptotic analysis of stochastic approximation algorithms for machine learning." Advances in neural information processing systems 24 (2011).
> > > [C] Lacoste-Julien, Simon, Mark Schmidt, and Francis Bach. "A simpler approach to obtaining an O (1/t) convergence rate for the projected stochastic subgradient method." arXiv preprint arXiv:1212.2002 (2012).
> > > [D] Nemirovski, Arkadi, et al. "Robust stochastic approximation approach to stochastic programming." SIAM Journal on optimization 19.4 (2009): 1574-1609.
> > > [E] Défossez, Alexandre, et al. "A simple convergence proof of adam and adagrad." arXiv preprint arXiv:2003.02395 (2020).

---

### Official Review · Reviewer_Z1Lm · 2022-07-14

**Rating:** 4
**Confidence:** 4
**Soundness:** 4 excellent
**Presentation:** 4 excellent
**Contribution:** 2 fair

**Summary:**

This work further investigates the use of stochastic Polyak stepsizes (SPS) first proposed in Loizou et al. (2021). The prior work had several limitations, essentially requiring that the global optimum is 0 (known) and that all minibatch samples are also minimized by the global minimizer. This work lifts this restriction and extends the analyses to the case where only a lower bound of the global optimum is known and when interpolation does not happen. The use of Polyak-type stepsizes in stochastic optimization is a relatively new direction, so many basic foundational questions about the setup need to be addressed. The discussion of section 4 provides insight into the bias in the SPS dynamics via a simple counter-example, and this nicely and clearly illustrates the difference between the SPS dynamics and that of the usual SGD dynamics. The newly proposed DecSPS is tested with some brief convex experiments, but no experiments on deep learning are done.

**Questions:**

In what way are the rates for DecSPS stronger compared to the prior rates on SGD?

**Limitations:**

.

**Strengths And Weaknesses:**

This work has some nice contributions, as it addresses some of the clear limitations of the prior SPS by Loizou et al. (2021). The line of research on SPS is new and, in my view, quite promising, so this work provides some clear value to the field

Overall, however, I feel that this work is a necessary, natural, but an incremental follow-up to Loizou et al. (2021). The analysis is not highly distinct or novel compared to the prior work. While there are certainly some new and interesting insights, the resultant rate is not really strong; under the interpolation assumptions, Loizou et al. (2021) established nice fast rates, but by lifting such assumptions, it seems that the rates are no longer surprising or impressive compared to the SGD rates. I found the results interesting, and I think this paper will serve as a good step forward in the SPS line of work, but I am not sure if these results are strong enough to be published in NeurIPS.

---

> ### Author Response · Authors · 2022-08-02
> **We are puzzled by your score --- did our best to reply to all your points**
>
> We thank the reviewer for their positive evaluation of our paper. Given your comments such as “clear value,” “necessary work,” “new interesting insights,” and “good step forward,” we are puzzled by your score. We did our best to understand and reply to your constructive criticisms, and we provide an answer to your concerns below:
>
> 1) **“Work is incremental, analysis not highly distinctive compared to Loizou et al.”**: our paper can certainly be thought of as a continuation of Loizou at al. (2021). As a result, some tools and proof techniques from the original paper can also be found here. However, this work studies a different problem: convergence to the exact solution under no interpolation. We proposed a new algorithm and combined the tools of Loizou with an Adagrad-like proof technique (this combination is novel). The combination of these techniques, along with the proof of bounded iterates, results in the first adaptive method without problem-dependent hyperparameters with convergence guarantees to the exact solution in convex stochastic programming.  Concretely, Theorem 5.2 and Proposition 5.4 present significant differences from the proof techniques of Loizou et al. (2021).
>
> 2) **“Resulting rate is not strong compared to Loizou et al.”**: We kindly disagree: the step-size in the original SPS paper behaves as a constant step-size, while the proposed DecSPS behaves as a decreasing step-size for classical SGD. This is the reason of the slower convergence. However, this modification is necessary in the non-interpolated setting for convergence to the exact solution. Similarly, for Vanilla SGD, the rates in stochastic convex programming (which also require a decreasing stepsize) are slower than the corresponding ones in the non-stochastic setting.
>
> 3) **“In what way are the rates for DecSPS stronger than the prior rates on SGD?”**. DecSPS is an adaptive method, hence the most natural comparison is with other adaptive methods like Adagrad and Adam. While convergence guarantees for Adagrad and Adam require bounded gradients/iterates, our method does not. Hence, it is stronger in the sense that the convergence guarantees require less restrictive assumptions. When comparing with SGD, convergence guarantees in the convex setting provide a similar asymptotic rate (see e.g. our reference [22]) but with a key difference: without knowledge of the gradient Lipschitz constant, the update of SGD cannot be guaranteed to reduce the loss at every step in the limit of vanishing noise. We will provide a comment on this in the potential camera-ready version of our paper.
>
> 4) **Results is not strong for NeurIPS?** We kindly ask the reviewer to engage with us for more details during the discussion period. As we highlight several times in our paper DecSPS is the first stochastic adaptive optimization method that converges to the exact solution without restrictive assumptions like bounded iterates/gradients. Only by this fact we believe our results is of great importance for the NeuriPS community. In addition, our proof techniques are novel as combined both SPS and Adagrad proof techniques that previously have not been used in the literature.
>
> We believe that all points raised by reviewer Z1Lm can be easily handled in the potential camera-ready version of our work. As we mentioned in our response, our algorithmic design and theoretical convergence guarantees are substantially different compared to the ones in Loizou et. al.  and provide new insights into the understanding of Polyak step-sizes. Let us highlight again that for strongly-convex optimization problems, DecSPS is the first stochastic adaptive optimization method that converges to the exact solution without restrictive assumptions like bounded iterates/gradients.
>
> If you agree that we managed to address all issues, please consider raising your mark. If you believe this is not the case, please let us know so that we have a chance to respond.

---

> > ### Comment · Reviewer_CpM5 · 2022-08-08
> > **Anwser to reviewer Z1Lm**
> >
> > Dear reviewer Z1Lm,
> >
> > I think it's a bit unfair to present this submission as "an incremental follow-up to Loizou et al. (2021)". Imo, this paper tries to a go a step further toward practical versions of adaptive methods exactly like SPS_max was introduced by Loizou et al (by adding a maximum threshold) when we move away from the interpolation setting.
> >
> > I support the authors on the fact that proper comparisons are to be done with other adaptive methods than SGD for above-mentioned reasons.
> >
> > Not only "the results are interesting", but I am sure the quality of the proof and the way they are compared with those of Loizou et al will be beneficial for the community working on SPS methods.
> >
> > I hope that the answer of the authors and my comment will make you considering an increase of your score.
> >
> > Do not hesitate to contact me directly for further discussions.

---

> > ### Comment · Reviewer_Z1Lm · 2022-08-09
> > **Thank you for your response**
> >
> > Thank you for your response. I believe that this paper makes useful and necessary contributions. I do not object to the other reviewers' majority view that the paper should be accepted.

---

> > > ### Author Response · Authors · 2022-08-09
> > > **Thank you**
> > >
> > > Thank you for your comment and for the positive evaluation of our paper! Since you believe it should be accepted, may you please consider increasing your score - this would totally boost our chances!
> > > Thank you again.

---

### Author Response · Authors · 2022-08-02
**Thanks to all reviewers**

Thanks to all reviewers for examining our manuscript and providing valuable suggestions. We are very glad to hear you found our paper interesting.
We take the reviewers’ criticism seriously, and we address all raised issues as individual comments. Here, we take the chance to summarize the main strengths of our contribution also using the reviewers' words.

To summarize, in this paper
1) We study the dynamics of stochastic Polyak stepsizes in **non-interpolated problems** and solve two practical issues with the SPS_max algorithm by Loizou at al. (2021) in non-interpolated problems: (a) the requirement for exact knowledge of the minima values f_B^*, for each batch B; and (b) lack of convergence to the exact solution.

2) We solved these problems by introducing a new stepsize: DecSPS, which combines the strong curvature adaptation properties of SPS_max while steadily decreasing over time. We provide strong convergence guarantees to the exact solution in non-interpolated (strongly) convex problems and show that other simple adaptations of SPS_max in this setting fail to converge to the exact solution. To the best of our knowledge, DecSPS is the first adaptive algorithm with convergence guarantees that do not require bounded gradients/iterates. This is the main finding of our paper.  We note that other adaptive methods (e.g. Adagrad and AMSgrad) instead require this assumption.

3) We validate our method thoroughly and compare against tuned versions of Adagrad, Adam, and RMSprop. The results show strong curvature adaptation and solid performance across the 3 considered experimental settings.

4) We discuss limitations and open directions for future research.

Reviewer Z1Lm found that our work “clearly addresses the limitations” of SPS_max by Loizou et al. (2021), that it provides “new and interesting insights,” and that it is a “nice contribution” with “clear value.” The reviewer also thinks this paper “will serve as a good step forward in the SPS line of work.” Reviewer 98CJ correctly recognized that ours is the first stochastic adaptive method that “converges to the exact solution without any restrictive assumptions such as bounded gradients.” This is indeed the main contribution and strength of our work. In addition, as reviewer 98CJ also writes, our algorithm is very easy to implement.
Reviewer 98CJ also thinks that “the paper is well written” and the “structure is easy to follow.” Reviewer CpM5 similarly writes that we provide a “good review of existing methods” and that “the paper is particularly well written, good intuition is given in the proofs which are easy to follow and makes the intentions clear to the reader.” We are glad for these comments since we took a lot of time polishing up our presentation.
Reviewer CpM5, who provided the highest score, also went through the proofs and found our theoretical guarantees “solid” and our problem of interest “well-posed.” On this note, Reviewer bV6Q also thinks our contributions are “clearly stated” and that the proofs are “well organized.”

*Main criticism: no deep learning experiments*. This is correct, yet at no point in our work we claim a contribution or focus on deep learning scenarios. We focus on convex non-interpolated problems and convergence to the exact solution, and we provide for the first time in the literature convergence guarantees for an adaptive method without the assumption of bounded iterates/gradients. Both our theory and experiments are restricted to the convex setting. In DL scenarios, convergence to a neighborhood is often satisfactory, or the interpolation condition is satisfied. In these scenarios, we would recommend implementing SPS of Loizou et al. (2021): the authors there discussed the performance and hyperparameter tuning in CNNs and ResNets and compared it with ADAM and its variants.

We hope you will engage with us in a back-and-forth discussion, and we will be most happy to answer any remaining questions.

---

### Meta-Review · Area_Chair_QnD8 · 2022-08-26

**Recommendation:** Accept
**Confidence:** Certain

**Metareview:**

As the reviewers have pointed out, it is a well-written paper with solid contribution.

**Award:**

No

---

### Decision · Program_Chairs · 2022-09-14

Accept